# Real-Time Fault Diagnosis for Hydraulic System Based on Multi-Sensor Convolutional Neural Network

**DOI:** 10.3390/s24020353

**Published:** 2024-01-07

**Authors:** Haohan Tao, Peng Jia, Xiangyu Wang, Liquan Wang

**Affiliations:** 1College of Mechanical and Electrical Engineering, Harbin Engineering University, Harbin 150000, China; taohaohan@hrbeu.edu.cn (H.T.); wangxiangyu325@126.com (X.W.); wangliquan@hrbeu.edu.cn (L.W.); 2Yantai Research Institute of Harbin Engineering University, Yantai 264000, China

**Keywords:** fault diagnosis, convolutional neural network (CNN), feature extraction, hydraulic system

## Abstract

This paper proposed a real-time fault diagnostic method for hydraulic systems using data collected from multiple sensors. The method is based on a proposed multi-sensor convolutional neural network (MS-CNN) that incorporates feature extraction, sensor selection, and fault diagnosis into an end-to-end model. Both the sensor selection process and fault diagnosis process are based on abstract fault-related features learned by a CNN deep learning model. Therefore, compared with the traditional sensor-and-feature selection method, the proposed MS-CNN can find the sensor channels containing higher-level fault-related features, which provides two advantages for diagnosis. First, the sensor selection can reduce the redundant information and improve the diagnostic performance of the model. Secondly, the reduced number of sensors simplifies the model, reducing communication burden and computational complexity. These two advantages make the MS-CNN suitable for real-time hydraulic system fault diagnosis, in which the multi-sensor feature extraction and the computation speed are both significant. The proposed MS-CNN approach is evaluated experimentally on an electric-hydraulic subsea control system test rig and an open-source dataset. The proposed method shows obvious superiority in terms of both diagnosis accuracy and computational speed when compared with traditional CNN models and other state-of-the-art multi-sensor diagnostic methods.

## 1. Introduction

Hydraulic systems are widely used in modern industry. In engineering applications, failures in hydraulic systems can cause enormous losses. Early stage fault detection and localization are essential to make hydraulic systems more reliable and efficient [1]. Intelligent fault diagnostic methods of hydraulic systems have been widely studied [2,3] overtime. Most of the hydraulic systems in practical applications are complex assemblies involving various mechanical, hydraulic, and electrical components. The coupling effects between the different components make real-time hydraulic system fault diagnosis a challenging task. To address this issue, data-driven fault diagnostic methods have attracted considerable attention from both industry and academia [4,5,6,7,8,9]. Models that are able to abstract valuable information from historical data are the key for these data-driven fault diagnostic methods [10], which have been widely investigated in recent years.

Machine learning (ML) models are used widely in data-driven fault diagnosis tasks. Many machine learning models, including support vector machines (SVMs) [7,11], neural networks (NNs) [4,5], random forest [12], and the extreme learning machine (ELM) [13], have been developed and applied to fault detection and classification. In these methods, the diagnostic accuracy is strongly dependent on the features that are fed to the classifier. To obtain more discriminative fault features from historical data, many signal processing methods, including the wavelet package transform (WPT) [9] and the fast Fourier transform (FFT) [12], have been used to extract signal features from the time or frequency domains [11]. These methods are usually combined with intelligent feature selection algorithms such as decision trees (DT), the binary particle swarm optimization (BPSO) algorithm [7], genetic algorithms (GAs) [12], and principal component analysis (PCA). In these ML-based methods, the final performance is affected strongly by feature extraction and selection, which are exhaustive processes that require expert knowledge of the target system, and they do not always generate the most discriminative features. These flaws all limit the generalization and accuracy of diagnostic methods based on ML.

As part of the rapid development of ML, deep learning (DL) [14,15,16] has shown enormous capabilities in the fault diagnosis field [17]. When compared with traditional ML models, DL models a have better nonlinear mapping ability, which allows them to learn more abstract and high-level fault features directly from raw signals automatically. Furthermore, many DL-based diagnostic models can integrate the feature extraction, feature selection, and fault classification processes to provide an end-to-end diagnosis model. Many new emerging DL techniques, including the deep belief network (DBN) [18], the deep stacked auto-encoder (DSAE) [19], the recurrent neural network (RNN) [20,21], the long-short-term-memory (LSTM) [22,23], and the convolutional neural network (CNN) [24,25], have been applied to fault diagnosis tasks and have demonstrated good performances when compared with traditional shadow ML methods. In hydraulic system, sensors of different types are used to collect different physical variables in various components. The DL-based models can extract fault-related features from the multi-sensor data through supervised learning. In [26], Huang et al. used a CNN model to extract fault features automatically from hydraulic sensor signals collected with different sampling rates. In [27], Zhang et al. applied deep transfer learning in the condition monitoring of a hydraulic system with multiple sensor signals. In [28], Zheng et al. proposed a LSTM combined with attention mechanism for tool condition monitoring. However, two issues significant for real-time diagnostic applications are neglected in these works. First, in hydraulic systems, not all the signal channels are related to fault. Using all channels of multi-sensors data will introduce redundant information, which will decrease the diagnosis performance of the model. Furthermore, from a computational perspective, a larger number of sensor channels implies a greater computational effort (increase in hardware costs, possibility of storage or physical space, computational and communication burden). 

To address these issues, in this paper, we propose a new DL-based diagnostic method that is more suitable for real-time diagnosis of hydraulic systems using multi-sensor data. The proposed method involves an improved CNN model named multi-sensor CNN (MS-CNN) that integrates the sensor selection and DL-based fault diagnosis. First, the MS-CNN is trained to extract the fault features from multi-sensor data. The learned features can be used for both sensor selection and fault diagnosis. Then, a random forest (RF)-based sensor selector is applied to select the most valuable and discriminative sensor channels using these learned features. Finally, the MS-CNN diagnostic model is fine-tuned with the signal data from the selected channels. This strategy reduces the redundant information and model complexity, which improves the computational speed and diagnostic accuracy significantly, making the model more generalizable and suitable for use in real-time diagnostic tasks. 

The main contributions made by this study can be summarized as follows: A new multi-sensor diagnostic model, termed a multi-sensor convolution neural network (MS-CNN), was proposed for hydraulic system diagnosis, which integrate a random forest (RF)-based sensors selector. This model can process multi-sensor data collected from different components to extract features for the purpose of both sensor selection and fault detection;A new MS-CNN-based sensor selection method was proposed to select fault-related sensor channels from multi-sensor data. The proposed method uses high-level features learned by a deep learning model for sensor selection. Compared to traditional signal-processing-based sensor-and-feature selection methods, the proposed method can find sensor channels containing higher-level fault features;A MS-CNN-based fault diagnostic method was proposed. The proposed method jointly integrates sensor selection, feature extraction, and fault diagnosis. It can find the sensor channels containing the most discriminative fault features through deep convolutional learning, which can improve the final diagnostic accuracy and reduce the communication burden and computational complexity in real-time diagnosis tasks. The experiments results show that the proposed MS-CNN-based fault diagnostic method offers significant improvements when compared with other state-of-the-art diagnostic methods in terms of both accuracy and computational speed.

The rest of this paper is organized as follows. Section 2 discusses related work, including other CNN-based diagnosis methods and existing sensor-and-feature selection methods. Section 3 introduces the framework of the proposed MS-CNN model and the diagnostic method based on it. Section 4 presents the results from testing of the proposed method on a public hydraulic dataset and a dataset collected from an electric-hydraulic system test rig. Conclusions are drawn and proposed future research areas are presented in Section 5.

## 2. Related Studies

In this section, we discuss previous studies on other CNN-based diagnoses and sensor-and-feature selection methods.

### 2.1. CNN Models Used in Diagnosis

The CNN is a special type of deep neural network (DNN) that uses convolution layers rather than fully connected layers for feature extraction. The convolution layer uses the local connection strategy and the weight sharing strategy to make the model more focused on local features and invariant to the location. Therefore, the CNNs are suitable for extracting features from structural data, including images and 1D signals. 

In fault diagnosis tasks, 1D CNNs are widely applied to mine the fault features from the collected time-domain sensor signals. In [22], a 1D CNN was used to extract hydraulic actuator fault features from 1D pressure signals, thus providing a huge advantage over other DL models (e.g., the DSAE and the LSTM). Jiang et al. [29] proposed a multi-scale CNN that made the model robust to the features of the various temporal scales in vibration signals. Huang et al. [26] proposed a modified 1D CNN that can extract features from signal data at multiple sampling rates. 

Different from 1D CNNs, 2D CNNs are more suitable to process 2D image data, but they can also be applied to signal-based diagnosis by transforming 1D signals into 2D images. For example, Wen et al. [30] converted 1D vibration signals into 2D gray-level images using a signal-to-image conversion approach. Similarly, a method was developed in [31] to convert 1D signals into a three-channel red-green-blue image. In [32], 1D signals were transformed into 2D matrices of wavelet packet coefficients and then used in CNN diagnosis models. 

The feature learning ability of the convolutional layers can also be combined with other deep-learning models. Sun et al. [33] used a backpropagation NN to learn filters that captured discriminative information through supervised training; these filters were then used to construct fault feature extraction CNNs. Guo et al. [34] proposed a CNN-based generative adversarial network (GAN) model to generate real damage data for diagnosis model training. In [35], a convolutional auto-encoder was applied to hydraulic system diagnosis. 

However, these CNN-based models lack sensor selection capabilities which limits the diagnostic performance and implies a greater computational effort when dealing with multi-sensor data composed of various physics variables of complex systems. In contrast, the proposed MS-CNN model integrates the process of sensor selection, feature extraction, and fault diagnosis. The model can automatically select the most discriminative and fault-related signal channels from multi-sensor data to improve the diagnostic accuracy. Meanwhile, reducing redundant information simplifies the model and thus makes it more suitable for real-time applications.

### 2.2. Sensor Selection and Feature Selection Methods

In order to reduce the redundant information and improve the diagnostic performance, it is necessary to find the sensor channels that contain the most discriminative and fault-related features. However, because of the high nonlinear and strong coupling relationships between the physical variables, finding the most valuable sensor channels or features for fault diagnostics is challenging. As a result, intelligent sensor selection and feature selection methods have therefore been widely studied. 

In [36], a PCA-based method was used to select sensors that best separated the healthy components from the faulty components to reduce the computational costs in real-time wind turbine diagnosis. Fawwaz et al. [23] proposed a GA-based feature selection method to select the most important features from various sensor data. Other selection algorithms, including decision trees [11] and the BPSO algorithm [7], were also studied in previous research. These selection methods are usually combined with ML-based diagnosis methods [7,8,9]. However, most of these selection methods use the low-level features (e.g., time-domain, frequency-domain) based on expert experience [9] and signal processing tools (e.g., WPT [7] and FFT [12]), or other methods such as frequency and mode shape assessment [37], and power spectral density of a vibration signal [38]. Therefore, the optimal sensor channels that contain discriminative high-level features may not be selected. 

Motivated by the nonlinear feature learning capabilities of deep learning models, this paper proposed a new MS-CNN-based sensor selection method for real-time diagnosis of hydraulic systems. In the proposed sensor selection method, the proposed MS-CNN diagnostic model is trained to extract the high-level fault-related features from multi-sensor data, which are used for both sensor selection and fault diagnosis. Then, a random forest (RF)-based sensor selector selects the most discriminative sensor channels using these learned features. The selected sensor channels are then used for the fine-tuning of the MS-CNN diagnostic model. Compared with the traditional sensor-and-feature selection method, the proposed sensor selection method can find the sensor channels containing higher level fault-related features, which can improve the final diagnostic accuracy.

## 3. Proposed Multi-Sensor-CNN Model

This paper focuses on real-time fault diagnosis of an electric-hydraulic system using multi-sensor data. In hydraulic systems, information contained in the single sensor data is usually insufficient to improve the diagnosis accuracy, because the features extracted from single sensor data are often subjected to variations in the operating conditions. Therefore, multi-sensor data that contain information of different physics variables have been widely utilized in data-driven diagnostic methods in recent studies. However, processing of multi-sensor data with numerous channels requires high computational power. In addition, not all the channels are valuable for diagnosis and the redundant channels will make this model more difficult to train and will decrease the diagnostic accuracy. To make use of the multi-sensor data more effectively, this paper proposes a multi-sensor CNN (MS-CNN) that can learn fault features for both sensor selection and fault detection.

### 3.1. Framework for Proposed MS-CNN Model

The framework for the proposed MS-CNN model is shown in Figure 1. The MS-CNN model consists of three main parts: a convolutional feature learning structure, a fault classifier, and a RF-based sensor selector. Furthermore, a random forest (RF)-based sensor selector is used to select fault-related sensor channels. 

#### 3.1.1. Convolutional Feature Learning Structure

The convolutional feature learning structure is used to extract abstract features from the sensor signal’s time series for fault classification and sensor selection. Similar to traditional CNNs, the convolutional feature learning structure of the MS-CNN is generally a cascaded structure composed of 1D convolutional layers and average pooling layers, followed by a global max pooling layer. The input format for the structure is a 2D matrix, formed by the 1D signal time series of multiple sensor channels. The main difference between the convolutional feature learning structure of MS-CNN and traditional CNNs is that there are two different 1D convolution layers that are used as the first layers of the convolutional feature learning structure at different stages, called the partial-input layer and the full-input layer; the input channel numbers for these layers are $N_{P}$ and $N_{F}$, (${N_{P}<N}_{F}$), respectively. Therefore, the input channel number for the MS-CNN is adaptive. By switching the structure of the first layer, the model can extract features from the signals from either all $N_{F}$ sensor channels or $N_{P}$ selected sensor channels. 

In a 1D convolutional layer, the input matrix $X={\left[x_{1},x_{2},\cdots ,x_{N}\right]}^{T}$ is formed by *N* channels. The 1D time series for each channel is ${x_{i}=\left[x_{i}(t)\right]}_{1\times L}\;$,where L is the length of the time series and i is the channel index. The input matrix is convolved with a group of filters $W_{j}={\left[W_{j}(i,k)\right]}_{N\times \delta }$ with a shape of (N×δ), where δ is the filter size. The layer contains *M* filters, and each filter outputs a transformed feature map that forms a channel in the output matrix *Z*, which is given as follows:$$Z={\left[z_{1},z_{2},\cdots ,z_{M}\right]}^{T}$$
(1)$$z_{j}(t)=\sigma \left[\sum_{i=1}^{N}\sum_{k=0}^{\delta -1}W_{j}(i,k)\cdot x_{i}(t+k)+b_{j}\right]$$
where $z_{j}$ is the $j^{th}$ transformed feature map and $z_{j}(t)$ is its value in the $t^{th}$ time step, $b_{j}$ is a bias term, and σ is a nonlinear activation function, which is a rectified linear unit (ReLU) function here. There are no padding methods in the convolutional layer, and thus the length of the output feature map is (L−δ+1).

Then, in the average pooling layer, the pooling operation is performed on the output matrix *Z*, and the output is the down-sampled matrix $\bar{Z}$, which is given as follows:$$\bar{Z}={\left[{\bar{z}}_{1},{\bar{z}}_{2},\cdots ,{\bar{z}}_{M}\right]}^{T}$$
(2)$${\bar{z}}_{j}(u)=Ave\left[z_{j}(\beta \cdot u:\beta \cdot (u+1))\right],\;\;u=1,2,\ldots ,Int\left[L/\beta \right]+1$$
where ${\bar{z}}_{j}$ is the $j^{th}$ channel of the pooled matrix, β is the pooling rate, and $Int\left[\cdot \right]$ is the integer ceiling function. For each time patch in the feature maps, the average pooling layer abstracts the average value to represent the entire patch. This operation can greatly reduce the number of feature dimensions while retaining the most important information to accelerate the calculation. Furthermore, the down-sampling characteristic of the pooling layers expands the receptive field of the next convolutional layer, which means that the model can extract the features on larger temporal scales. 

The $\bar{Z}$ matrix is then fed into the next convolutional and pooling layer pair, where the multilayer nonlinear transformation of this cascaded learning structure allows the model to extract more abstract high-level features. After the last convolutional layer, a global max pooling layer was used to extract the global maximum value of each feature map; the output of this layer is a feature vector that contains the most representative features of the input multi-sensor signals. The feature vector P is represented by the following:$$p_{j}=MAX\left[{\bar{z}}_{j}(1:L^{\prime })\right]$$
(3)$$P={\left[p_{1},\ldots ,p_{j},\ldots ,p_{M^{\prime }}\right]}^{T}=\left[\begin{array}{c} MAX\left[{\bar{z}}_{1}(1:L^{\prime })\right]\\ \ldots \\ MAX\left[{\bar{z}}_{M^{\prime }}(1:L^{\prime })\right] \end{array}\right]$$
where ${\bar{z}}_{j}$ is the $j^{th}$ channel of the output matrix of the last convolutional layer $\bar{Z}$, and $L^{\prime }$ and $M^{\prime }$ are the length and the channel number of the output matrix of $\bar{Z}$, respectively.

The model can extract the corresponding feature vector from the input multi-sensor signal matrix X. The feature vector of the sample can also be represented by the following:(4)$$P=\left\{\begin{array}{cc} G(X) & ,X={\left[x_{1},x_{2},\ldots ,x_{N_{P}}\right]}^{T}\\ \tilde{G}(X) & ,X={\left[x_{1},x_{2},\ldots ,x_{N_{F}}\right]}^{T} \end{array}\right.$$
where G(X) represents the mapping relationship of the partial-input layer-based model, $\tilde{G}(X)$ represents the mapping relationship of the full-input layer-based model, and $x_{i}$ is the $i^{th}$ channel of the input matrix.

#### 3.1.2. Fault Classifier

The fault classifier is used to predict the fault conditions with the extracted features. The classifier only contains a fully connected (FC) layer with a soft-max activation function. The FC layer transforms the obtained feature vector P into a multi-class probability vector that indicates the predicted health conditions of the system. The probability vector is calculated as follows: (5)$$Q=softmax(\omega P^{T}+b)$$
where $Q={\left[q_{i}\right]}_{1\times C}$ is the probability vector of the shape (1×C), $q_{i}$ is the predicted probability value of each class for the input sample, D is the number of labels, ω is the weight matrix of the shape $\left(M^{\prime }\times C\right)$, b is the bias vector, and softmax(·) is the softmax active function, which can be expressed as the following:(6)$$softmax({\left[z_{1},\ldots ,z_{C}\right]}^{T})={\left[\frac{e^{z_{1}}}{\sum_{k=1}^{C}e^{z_{k}}},\ldots ,\frac{e^{z_{C}}}{\sum_{k=1}^{C}e^{z_{k}}}\right]}^{T}$$

Note that a dropout operation with a drop rate of 20% is used in the FC layer. In each training epoch, 20% of the nodes in the multi-sensor feature vector are selected randomly and set to zero. This operation can prevent overfitting and increases the training speed. 

To train the diagnosis model, we feed the training samples into the model and acquire the corresponding predicted probability vectors. Then, the binary cross-entropy loss between the probability vectors and the label vectors of all the training samples in a training batch is calculated as the loss function, which is represented by the following:(7)$${Loss}_{BCE}=\frac{1}{C\cdot N}\sum_{n=1}^{N}\sum_{k=1}^{C}l_{k}^{\left(n\right)}$$
(8)$$l_{k}^{\left(n\right)}=-\left[y_{k}^{\left(n\right)}log{(q}_{k}^{\left(n\right)})+(1-y_{k}^{\left(n\right)})log(1-q_{k}^{\left(n\right)})\right]$$
where N is the number of samples in a training batch, and $y_{k}^{\left(n\right)}$, $q_{k}^{\left(n\right)}$, and $l_{k}^{\left(n\right)}$ are the training label, the predicted probability, and the corresponding loss of the $k^{th}$ label and the $n^{th}$ sample, respectively.

In each training epoch, the loss is backpropagated to the weights and biases in the FC layers and the convolutional layers. The Adam optimizer then adjusts these parameters to minimize the loss. Through an epoch-wise parameter update, the model is trained to extract fault-related features from the input multi-sensor signal matrix and use them to classify the system’s health conditions. 

### 3.2. Training of MS-CNN

The MS-CNN training process can be divided into three stages: the pre-training stage, the sensor selection stage, and the fine-tuning stage. The pre-training stage gives the model its primary ability to extract the fault features required for fault classification and sensor selection. At this stage, the model is trained on the pre-training dataset, which is a randomly selected subset of the training dataset. The full-input layer is used as the first layer of the model during this stage, and the model thus takes all $N_{F}$ sensor channels as its input. For each sample, the input matrix is given by the following:(9)$$X^{\left(n\right)}={\left[\varphi_{1}^{\left(n\right)},\varphi_{2}^{\left(n\right)},\ldots ,\varphi_{N_{F}}^{\left(n\right)}\right]}^{T}$$
where $\varphi_{i}^{\left(n\right)}{=\left[\varphi_{i}^{\left(n\right)}(t)\right]}_{1\times L}$ is the signal time series of the $i^{th}$ sensor channel and n is the sample index.

During the supervised pre-training, the model can learn fault-related features from all sensor channels automatically. Values in vector P are the representation of the fault-related information in sensor signals. We proposed a sensor selection method based on the feature vectors to find the sensor channels that contain the most discriminative fault-related features. 

After pre-training, the sensor selection operation is performed. During the sensor selection stage, the well pre-trained model with the full-input layer is used to select the optimal sensor channels required for further training and testing. First, we create a single-sensor signal matrix composed of one signal time series and another $\left(N_{F}-1\right)$ zero time series for each sensor channel. The new single-sensor signal matrices have the same shape as the $N_{F}$-channel multi-sensor signal matrix, and these new single-sensor signal matrices can be given as the following:(10)$$V^{\left(n)(i\right)}={\left[O,O,\ldots ,\varphi_{i}^{\left(n\right)},\ldots ,O\right]}^{T}$$
where $V^{\left(n)(i\right)}$ is the single-sensor signal matrix of the $n^{th}$ sample and the $i^{th}$ sensor channel. The $i^{th}$ channel of the matrix $V^{\left(n)(i\right)}$ is the same as the $i^{th}$ channel of the multi-sensor signal matrix $X^{\left(n\right)}$, while the other channels are zero time series.

The feature vector of each sensor is $X^{\left(n)(i\right)}={\left\{x_{j}^{\left(n)(i\right)}\right\}}_{N_{D}\times 1}$, where $N_{D}$ is the output channel number of the last convolutional layer, it can be denoted as the following:(11)$$X^{\left(n)(i\right)}=\tilde{G}(V^{\left(n)(i\right)})$$
where $\tilde{G}(\cdot )$ represents the mapping relationship of the full-input layer-based model, which outputs the feature vector of the input single-sensor signal matrix. The feature vector $X^{\left(n)(i\right)}$ only relates to sensor signal $\varphi_{i}^{\left(n\right)}$, and it is the representation of the fault-related information in its corresponding sensor channel. Therefore, the features in vector $X^{\left(n)(i\right)}$ can be used for sensor selection. 

The global feature vector is denoted as the follows:(12)$${\overset{ˇ}{X}}^{\left(n\right)}=\left\{x_{1}^{\left(n)(1\right)},\ldots ,x_{N_{D}}^{\left(n)(1\right)},\ldots ,x_{N_{D}}^{\left(n)(N_{F}\right)}\right\}$$

Then, a random forest classifier is trained on these features and gains the mean decrease in Gini impurity (*MDG*) of each feature, which can be denoted as
(13)$$MDG\left(x_{j}^{\left(n)(i\right)}\right)=\frac{1}{M}\sum_{t\in T}\left[GI\left(A\right)-\frac{n_{L}}{n}GI\left(A_{L}\right)-\frac{n_{R}}{n}GI\left(A_{R}\right)\right]$$
where A is the collection of observations falling in node t, $A_{L}$, and $A_{R}$ are the observations collections of its left and right child nodes, respectively. n, $n_{L}$, and $n_{R}$ are the number of observations in A, $A_{L}$, and $A_{R}$ respectively. T is the collection of nodes using feature $p_{j}^{\left(n)(i\right)}$ in the random forest classifier. M is the number of trees in the random forest classifier. $GI\left(A\right)$ is the Gini impurity of A, which is calculated as follows:(14)$$GI\left(A\right)=\sum_{k=1}^{N_{classes}}p_{k}\left(1-p_{k}\right)$$
where $p_{k}$ is the proportion of class k in collection A, and $N_{classes}$ is the number of classes in collection A.

The corresponding feature score of each sensor channel is calculated as follows:(15)$$F^{\left(i\right)}=\frac{1}{N}\sum_{1\leq n\leq N}\sum_{1\leq j\leq N_{D}}MDG\left(x_{j}^{\left(n)(i\right)}\right)$$
where N is the sample number in the training dataset. This metric measures the forest-wide contribution of the sensor channel at separating the different classes and constitutes one measure of sensor importance. The eight sensor channels with the highest feature scores are selected. In order to further reduce the redundant information, the Pearson correlation coefficient is used to measure the correlation between selected sensor channels, which is calculated follows:(16)$$\rho \left(x^{\left(i\right)},x^{\left(j\right)}\right)=\frac{cov\left(\sigma_{x^{\left(i\right)}},\sigma_{x^{\left(j\right)}}\right)}{\sigma_{x^{\left(i\right)}}\sigma_{x^{\left(j\right)}}}$$

The sensors with high absolute values of correlation coefficients are screened because they contain similar information. After sensor selection, the number of sensor channels is reduced from $N_{F}$ to $N_{P}$.

Next, the selected sensor channels are used as the input channels for the partial-input layer for further fine-tuning and testing. Sensor channels with higher feature scores are more likely to contain more discriminating fault features. Therefore, when compared with the channels with lower feature scores, the selected channels are considered to be more relevant to the fault conditions, making them more suitable for further training and online diagnosis. 

To accelerate convergence and reduce the computational costs during the pre-training stage, the pre-training dataset only takes a small proportion of the entire training dataset so that it does not have a sufficient sample volume to allow the model to achieve satisfactory diagnostic results. Therefore, the model should be further fine-tuned on the training dataset in the fine-tuning training stage. During the fine-tuning training stage, the partial-input-layer is used as the first layer of the model, and the input multi-sensor signal matrix, which is formed by the selected channels, is given by the following:(17)$$X^{\left(n\right)}={\left[{\bar{\varphi }}_{1}^{\left(n\right)},{\bar{\varphi }}_{2}^{\left(n\right)},\ldots ,{\bar{\varphi }}_{N_{P}}^{\left(n\right)}\right]}^{T}$$

Because only the data from $N_{P}$ selected sensor channels, rather than from all channels, are selected as inputs to the model during both the fine-tuning and online diagnosis stages, the model complexity and the computational cost are greatly reduced. More importantly, the sensor selection operation ensures that the model is trained on optimal sensor channels that contain more features that contribute to the prediction accuracy, which accelerates the model convergence and increases training efficiency. 

### 3.3. MS-CNN-Based Fault Diagnosis Method

The framework of the proposed fault diagnostic method is shown in Figure 2. This framework is built to detect faults in an electric-hydraulic system in real time using raw multi-sensor data signals. The process to establish this framework can generally be described as follows. 

The multi-sensor data and the corresponding working conditions are acquired from the historical data and the records of the target hydraulic system. Multi-sensor data are split into 400 s long time segments that are used as training and testing samples while the corresponding working conditions are used as labels for these samples. The training dataset and the testing dataset are selected randomly from these labeled samples. The MS-CNN-based diagnostic model is constructed and trained on the training dataset. The training stage contains the processes of pre-training, sensor selection, and fine-tuning, and is performed offline. During training, the optimal sensor channels are selected from all channels. After training, the model will be tested using the test dataset to assess the diagnosis accuracy and then used for real-time diagnosis. 

During the real-time diagnosis stage, the sensor signals are collected from the target hydraulic system in real time. Diagnosis is performed every 10 s to predict the system’s health condition periodically. The input of the model is the matrix of 400 s long signals from the selected optimal channels, the format of which is the same as the training samples. 

Similar to the proposed MS-CNN, in [22,26], the 1D signals from different sensors are channel-wise packaged to form the input matrix for a deep learning-based diagnosis model which also enable the model to take advantage of multi-sensor data. However, in these studies, sensor selection was neglected, which will introduce redundant information into the deep learning model and limit the diagnostic performance. In contrast, the MS-CNN proposed in this paper eliminates the redundant sensors automatically and simplifies the model. Using this simpler model, the training and online diagnosis process can be performed rapidly. In addition, monitoring of the eliminated sensors can be conducted at a lower sampling frequency, which also reduces the communication and data storage costs. All these advantages make the proposed method more suitable for use in real-time diagnosis applications. 

## 4. Case Study

This section presents the results from testing of the proposed method on a public hydraulic dataset and a dataset collected from an electric-hydraulic system test rig. 

### 4.1. Case I

The proposed real-time diagnosis method was examined in an electric-hydraulic subsea control system (EHSCS) [39], which is a typical hydraulic system used in subsea oil and gas production.

#### 4.1.1. Test Rig

An EHSCS test rig was built to collect the multi-sensor data from the system. The test rig framework is shown in Figure 3. The test rig mainly consists of a hydraulic power unit (HPU) with two hydraulic pumps (MP1–2), an umbilical emulator, a subsea control module (SCM), and a hydraulic test station (HTS). The HTS contains a series of hydraulic quick couplings (QC1–4), a loading system (MP3), and a group of valves that emulate faults in the hydraulic components. The SCM mainly contains a hydraulic accumulator (A1), a hydraulic filter (F3), and a DCV (V6). Note that in a real EHSCS, the hydraulic accumulators of all the SCMs are connected together, and thus the influence of the other SCMs is emulated using an accumulator (A2) connected through long pipes. There are 20 sensors mounted on the test rig and the sensor types are described in Table 1. A programmable logic controller (PLC) controls the DCV to operate the actuator and collects the multi-sensor signals used as training and testing samples. Each sample is a 20-channel signal segment of 400 s, and the sampling frequency is 10 Hz. Therefore, each sample contains 20×10×400=80,000 data points. The signal in each sample covers a time period that contains an open-and-close cycle of the actuator and an HPU switching operation, which is suitable for emulation of a typical operation in subsea actuators. 

In this case, nine types of fault were injected into the different components shown in Table 2, and thus 10 health conditions (including the normal condition) were studied in total. Samples with different faults were collected under various operating conditions. The loading system pressure was set to values of 2,6,10,14, and 18 MPa and the HPU pressure was set to values of 13,14,15, and 16 MPa, and thus 4×5=20 operating conditions were emulated. For each operating condition and fault condition, we collected 10 samples, and thus 20×10×10=2000 samples were collected in total. Figure 4 displays the pressure sensor signal of PS3 and PS6 under normal situations and a fault of BLK2, with a loading system value of 2 MPa; the signals of a normal system with a loading system value of 14 MPa are also shown in Figure 4. It could be seen that the waveform of sensor signals can be significantly influenced by faults in the system, as well as in the operation conditions.

To evaluate the effectiveness of the proposed method, a 10-fold cross-validation [15] was conducted in this study. The collected samples were divided randomly into 10 groups. Each group was selected as the test set in turn, while the remaining groups were used as the training set. The average value and the standard deviation of the accuracy over all classes in these 10 tests were used to evaluate the method’s diagnosis performance. Further comparisons were also performed on these terms. 

The calculation environment used for the MS-CNN model and for other algorithms for comparison in the experiments is as follows: Python 3.6, Pytorch; computer operating system: Windows 10; a CPU frequency of 3.3 GHz, 8192 MB of RAM, and an NVIDIA RTX3060 GPU.

The MS-CNN is composed of three convolution layers, two average pooling layers, and a fully connected layer. The kernel size of the filter is 29 and the pooling rate is nine. The configuration of the MS-CNN is shown in Table 3. During training, all CNN models shared the same training parameters. The models were trained using the Adam optimizer with a step decay in the learning rate. The initial learning rate was set at 0.0005, and the decay was 0.98. The training was terminated when the training reached the 300th epoch. Both the model structure and the training hyper-parameters are predefined considering the test performance and the computational cost of the experiment platform.

#### 4.1.2. Comparison of Results with Other Sensor and Feature Selections Based Methods

The feature extraction ability of the proposed MS-CNN was evaluated through comparison with existing data-driven diagnosis methods based on sensor selection and feature selection algorithms, including PCA [36], DT [11], and GA [23]. The features used in these methods were extracted from all sensor channels, including the time-domain features [7,9,11], FFT-based features [12], and WPT-based features [8], which are very commonly used in data-driven diagnosis methods. For further comparison, an artificial neural network (ANN) diagnosis model and a random forest (RF)-based diagnosis model using all these manual features are considered during this comparison. All methods were trained on the same dataset for fairness. 

The overall accuracy for each method is listed in Table 4 and a receiver operating-characteristic (ROC) plot is shown in Figure 5. When compared with the other methods, the proposed method obtained higher diagnosis accuracy, which ranged as high as 99.40%. In addition, the MS-CNN’s standard deviation was also clearly lower than that of the other methods, which illustrates that the diagnostic results of the MS-CNN are more reliable. The steepness and areas of the ROC curves also demonstrate the discrimination characteristics of the MS-CNN. Among all the sensor-and-feature selection-based methods, the DT+ANN method achieved the best performance in terms of mean diagnostic accuracy, with a mean accuracy of 94.10%, which means that the DT algorithm can find the most valuable and discriminative features from all manually extracted features for diagnosis. However, the pure sensor-and-feature selection methods can only improve the performance slightly because when compared with the features learned by the MS-CNN, these manually designed features are often subject to variations in the operating conditions and are not sufficiently discriminative for systems with various loading conditions.

To provide a better visualization of the feature extraction capability of the proposed method, we used the t-distributed stochastic neighbor embedding (t-SNE) technique to reduce the dimension of the MS-CNN-extracted feature vector and the features selected by the comparison methods and then plotted them in 2D maps. The results are shown in Figure 6. Samples with different labels are indicated by different colors. The plots show that the features extracted using the proposed method are more clearly separated when compared with the other methods’ selected features, which verifies that the features extracted by the proposed approach are more discriminative and robust to variations in the operating conditions. 

#### 4.1.3. Comparison with Other Multi-Sensor Fusion Methods for the CNN Model

To illustrate the effectiveness of the proposed sensor selection ability based on high-level features, the diagnostic accuracy and computation time of the MS-CNN are compared with traditional CNN using all sensor channels in this section. In [22], two kinds of multi-sensor data fusion methods for CNNs were studied, in which the multiple signal segments were transformed into a long 1D segment in a time domain and a 2D matrix channel-by-channel, respectively. Inspired by this work, we built CNN models using these two transformation methods for comparison, which we named CNN-1 and CNN-2, respectively. A comparison of CNN-1, CNN-2, and MS-CNN was conducted in terms of the 10-fold cross-validation accuracy and average computation time during the training and 200-sample testing stages. Comparison results are shown in Table 5. The hyper-parameters of these models are the same except for the input channel numbers. 

The overall testing accuracies of the models are 99.40% (MS-CNN), 89.55% (CNN-1), and 98.30% (CNN-2). The MS-CNN performs better than CNN-1 and CNN-2 in terms of both average accuracy and standard deviations. It can be concluded that the features extracted from the MS-CNN-selected sensors are more discriminative and robust than the features that were extracted from all channels, and thus the proposed MS-CNN is proved to be able to find the most fault-related sensor channels effectively from the multi-sensor data. 

Furthermore, the time costs during both the training stage and the testing stage for the MS-CNN are significantly lower than those of the other models because the MS-CNN only uses the most valuable channels as inputs, which reduces the computational and storage costs of forward propagation. On the one hand, the MS-CNN’s faster testing speed makes it more suitable for time-sensitive applications including real-time fault monitoring. On the other hand, the sensor selection strategy not only accelerates the computation speed for each training epoch, but it also improves the algorithm’s convergence speed. 

### 4.2. Case II

To further validate the effectiveness of the proposed method, an experiment is conducted on an open-source dataset.

#### 4.2.1. Experiment Description

The experimental data derives from an open-source dataset detailed in [40], which is also used to validate other state-of-the-art diagnostic methods in [26,27,41,42]. In this dataset, multi-sensor data are collected from the test rig shown in Figure 7. Several kinds of faults are simulated in the test rig, which differ in fault types, severity, and duration. The components and simulated fault conditions are presented in Table 6. The sensors monitoring the system are shown in Table 7.

In this section, the fault diagnosis tasks for each component are conducted, respectively, and the severity of the failure is used as the label in each case. For example, in the diagnosis of the cooler C1, the label can be denoted as Y∈{full efficiency, reduced efficiency, close to total failure} = {[1,0,0], [0,1,0], [0,0,1]}. Since the sensor channels in multi-sensor data are collected with different sampling rates, an unsampling operation is conducted on signals with sampling rates of 1 Hz and 10 Hz in order to make sure that signals in all channels have the same length. 

In this study, a 5-fold cross validation was performed, similar to the experiment in [26]. The average value and the standard deviation of the accuracy over all classes were used to evaluate the method’s diagnosis performance. Further comparisons were also performed on these terms. 

#### 4.2.2. Discussion of Results and Comparison of Performance with Other State-of-the-Art Methods

The configuration of the MS-CNN is the same as in the former experiment. First, to verify the effectiveness of the sensor selection, a traditional CNN method was used as the baseline method for comparison. Here, the hyper-parameters of the traditional CNN model are the same as the MS-CNN with the fully connected layer, and it uses all channels in the multi-sensor data as the input. In addition, several state-of-the-art intelligent diagnostic methods were also introduced for comparison. The comparison results are shown in Table 8.

In general, in fault detection of all components, the proposed model achieves the best diagnostic performance compared to state-of-the-art published work in recent years based on this dataset. In [41], Liu et al. extracted a series of time and frequency domain fault features for all sensor channels, then, a multi-channel data selection method based on linear discriminant analysis (LDA) was used to screen out sensitive features for a hybrid kernel extreme learning machine (HKELM) fault detection model. A similar feature extraction and selection process can also be found in [40]. Similar to LDA, in [42], Wang et al. used the Pearson correlation coefficient (PCC) and neighborhood component analysis (NCA) to select data channels for a heterogeneous ensemble deep neural networks diagnostic model based on a stacked sparse auto-encoder (SSAE) and a deep hierarchical extreme learning machine (HELM). In the studies mentioned above, artificially designed features are used to represent information from multiple sensors; thus, the feature selection and diagnosis rely heavily on expert knowledge. In contrast, in our proposed method, manual feature extraction is not required. The CNN model can automatically learn discriminative, robust, and higher-level features from raw signals of the multi-sensor data, which improves the diagnostic performance in applications significantly as shown in Table 8. 

In [27], Zhang et al. transfer a pretrained natural language processing (NLP) model GPT-2 for a fault diagnosis task using multi-sensor signal data. The self-attention mechanism enables the GPT-2 model to identify the critical fault-related sensor channels, which improves the effectiveness of handling the multi-sensor data with large sensor numbers. However, the GPT-2 is a complex model. Such a model requires a large computational cost, which makes it difficult to apply to real-time diagnosis. Additionally, compared to the GPT-2 that was developed for NLP tasks, our proposed CNN-based model can be focused more on local features, which makes it more suitable to deal with data with spatial or temporal information such as a signal time series. The result shows that the proposed MS-CNN outperformed the GPT-2 when facing faults in the accumulator and pump. 

In [26], Huang et al. used the Pearson correlation coefficient to select fault-related sensor channels and reduce the redundant information. Then, they used a CNN with multiple independent parallels to extract features for sensors with different sampling rates. However, the sensor selection process and feature extraction process are separated in this framework. The PCC only reflects the linear correlation between signals and fault labels; thus, the sensor channels selected may not contain the higher-level and nonlinear fault-related features. In the proposed MS-CNN, the sensor selection is based on the features learned through supervised training, and the RF-based sensor selector can find out the sensor channels containing higher-level fault-related features, which can improve the final diagnostic accuracy as shown in Table 8.

The results show that the proposed MS-CNN outperformed the traditional CNN using all channels in multi-sensor data in the diagnosis of all components, especially the accumulator and pump, which illustrates that the sensor selection of the proposed MS-CNN can reduce the redundant information of the multi-sensor data and therefore improve the diagnostic performance of the model. To better illustrate this, we used the t-SNE technique to visualize the feature vectors extracted with the MS-CNN and the traditional CNN in the diagnosis of different components. The results are shown in Figure 8. Samples with different labels are indicated by different colors. The plots show that the features extracted with the MS-CNN are more clearly separated when compared with the features extracted with the traditional CNN, which verifies that the sensor channels selected by the MS-CNN contain the most discriminative fault-related features. 

The sensor channels selected by the MS-CNN for each fault are shown in Table 9. These optimal sensor channels are selected according to their sensitivity to specific faults and their robustness to varying operation conditions, which depend on the dynamic characteristics of hydraulic components and the installation positions of the sensors. Therefore, the optimal sensor channels differ in different hydraulic systems and faults; the MS-CNN can automatically select the most discriminative and robust sensor channels for diagnosis.

## 5. Conclusions

The main contribution of this research is the development of a fault diagnosis method based on the MS-CNN that incorporates feature extraction, sensor selection, and fault diagnosis into an end-to-end model. Compared with other fault diagnostic methods based on manually designed fault features, the proposed model takes advantage of the abstract features extraction ability of the deep learning model, which makes it able to learn fault features automatically from the raw signals without any signal processing operations. The comparison of the diagnostic results shows that the proposed MS-CNN outperformed those low-level features-based methods significantly. That is because the manually designed and signal processing-based fault features are usually subject to the variation in operation conditions and signal noise. In contrast, the proposed MS-CNN can learn more robust and comprehensive features though supervised learning. 

In contrast to the traditional CNN diagnostic model, the MS-CNN integrates the proposed sensor selection in the learning structure. The comparison results between the MS-CNN and traditional CNN diagnostic models has illustrated that this strategy can effectively reduce the redundant information of the multi-sensor data to improve the diagnostic accuracy. The reduction in sensor channels also decreases the cost in calculation, communication, and data storage; all these advantages makes the MS-CNN more suitable for the real-time diagnosis of systems with data from multiple sensors. 

Moreover, the proposed sensor selection method is based on the higher-level features learned from the deep conventional structure instead of from manually designed or signal processing-based features. The result of the experiment conducted on the public hydraulic dataset shows that the proposed method is more accurate in the fault detection of all components compared with other deep learning diagnostic methods combined with sensor selection methods such as PCC, LDA, and NCA. 

The proposed MS-CNN model provides a potential method for use in various fault diagnosis tasks in other complex systems with multiple sensors. Of course, several points need to be pointed out as further research directions. First of all, since the labeled fault samples are difficult to obtain in SCSs and many other industrial applications, it is necessary to investigate the problem of insufficient diagnostic accuracy while, at the same time, lacking labeled training samples. Secondly, the proposed sensor selection can be combined with other kinds of deep learning models such as self-attention-based models for the purpose of improving the efficiency of feature extraction for multi-sensor data. 

## Figures and Tables

**Figure 1 sensors-24-00353-f001:**
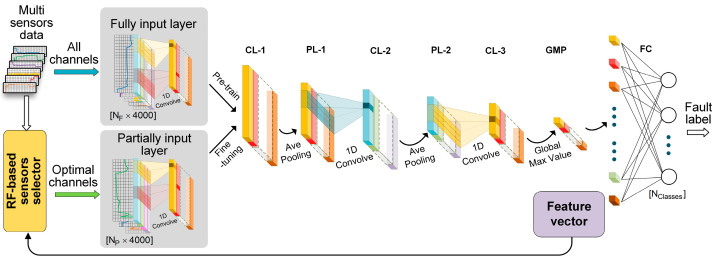
Framework of the proposed multi-sensors-CNN model, in which “CL-1”, “CL-2”, and “CL-3” are the convolutional layers, “PL-1” and ”PL-2” is the average pooling layers, “GMP” is the global max pooling layer, and “FC” is the fully connected layer.

**Figure 2 sensors-24-00353-f002:**
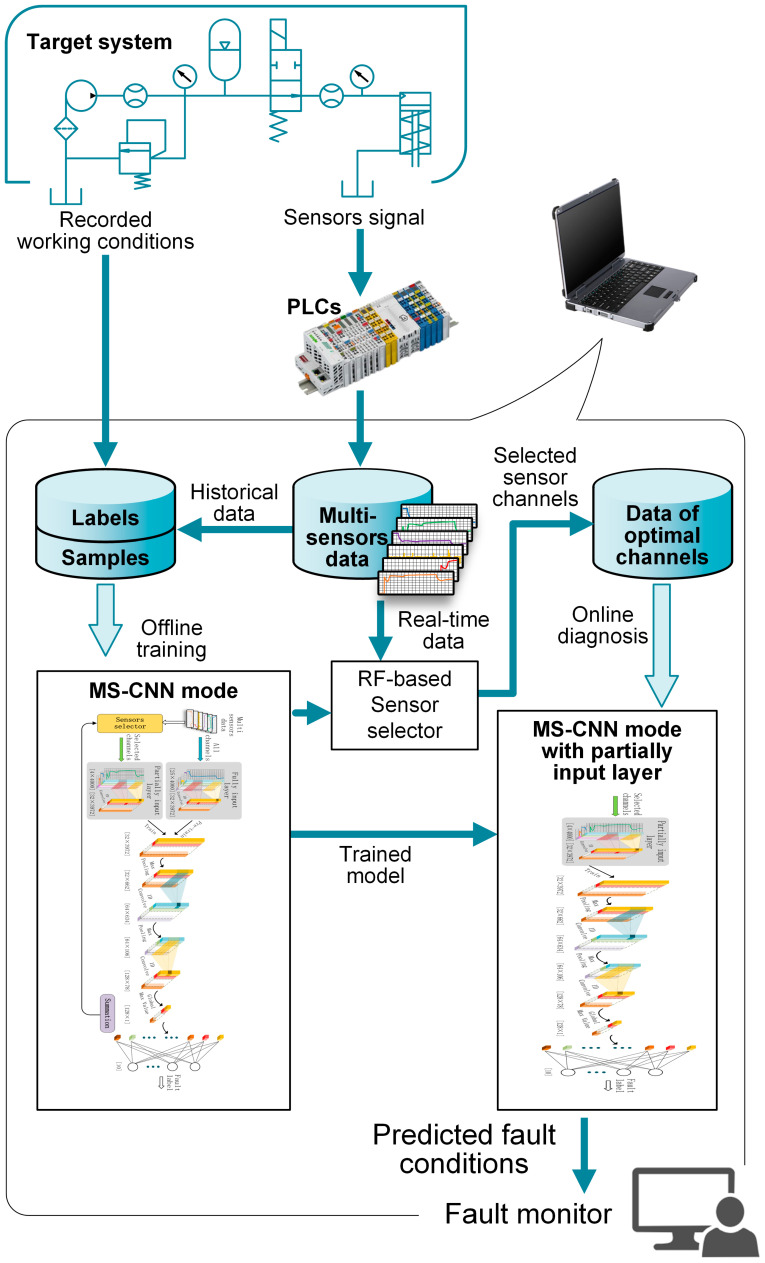
Framework of proposed fault diagnosis method based on the MS-CNN.

**Figure 3 sensors-24-00353-f003:**
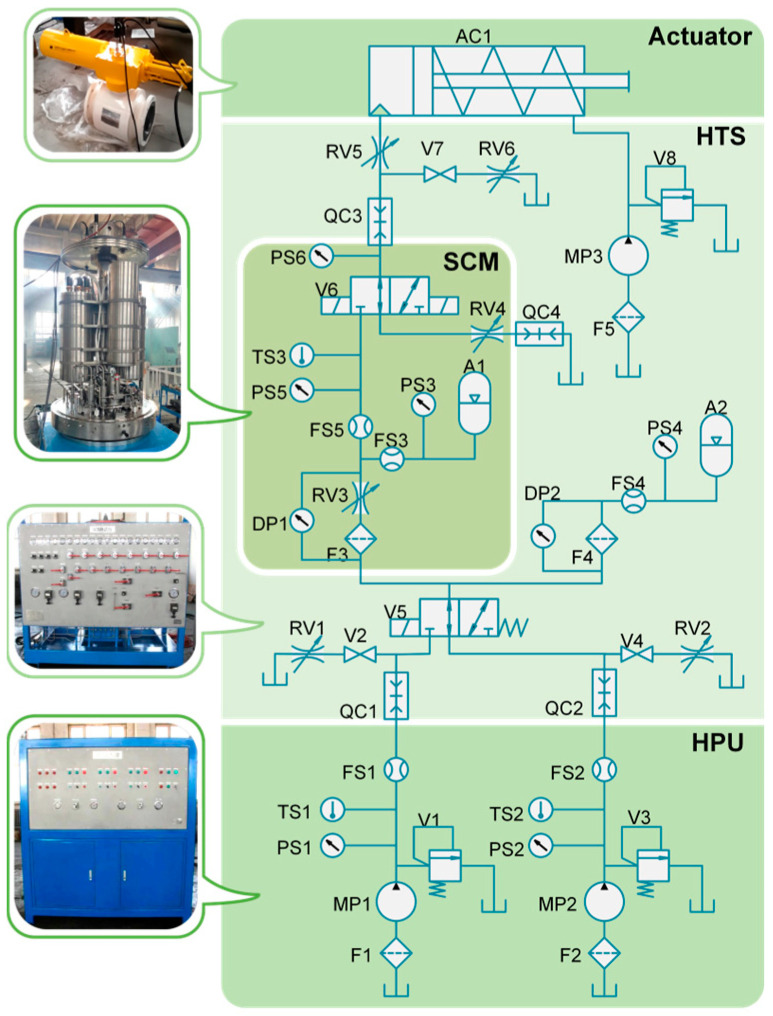
Schematic of the EHSCS test rig setup.

**Figure 4 sensors-24-00353-f004:**
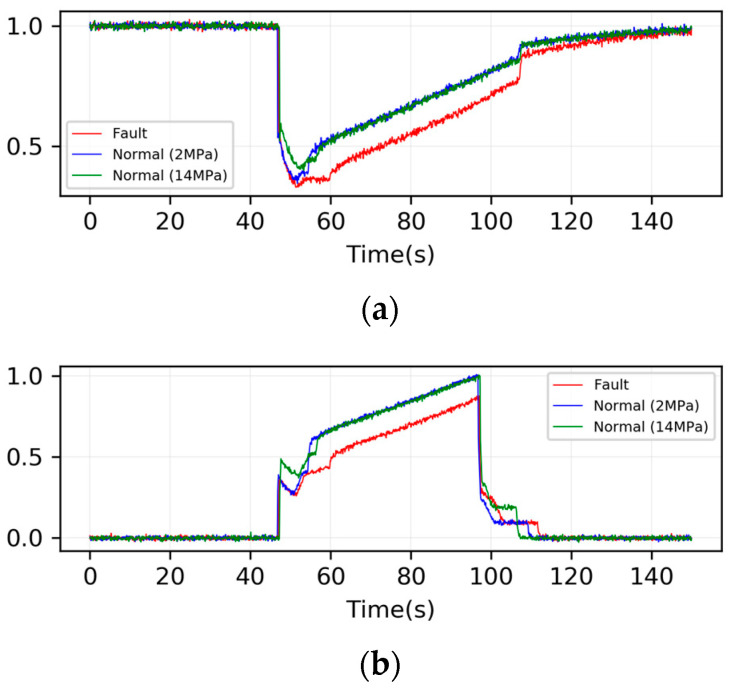
Sensor signals collected from test rig (**a**) PS3 and (**b**) PS6.

**Figure 5 sensors-24-00353-f005:**
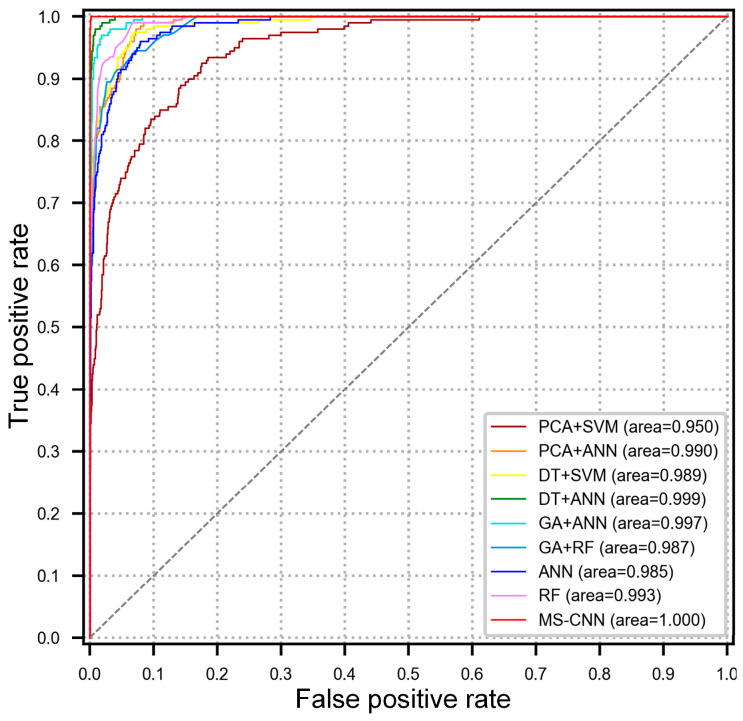
ROC curves for the MS-CNN and other methods.

**Figure 6 sensors-24-00353-f006:**
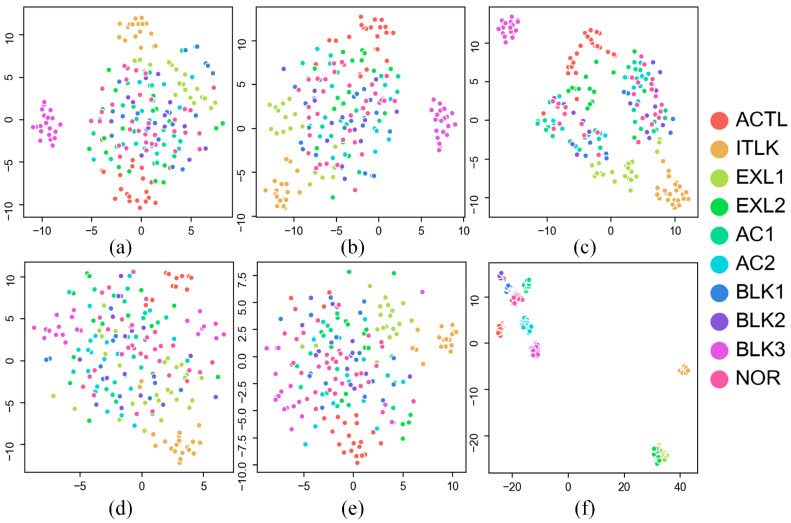
Feature visualization via t−SNE reduced from fault features for the testing dataset with (**a**) all manual features, (**b**) PCA, (**c**) DT, (**d**) GA + ANN, (**e**) GA + RF, and (**f**) MS−CNN.

**Figure 7 sensors-24-00353-f007:**
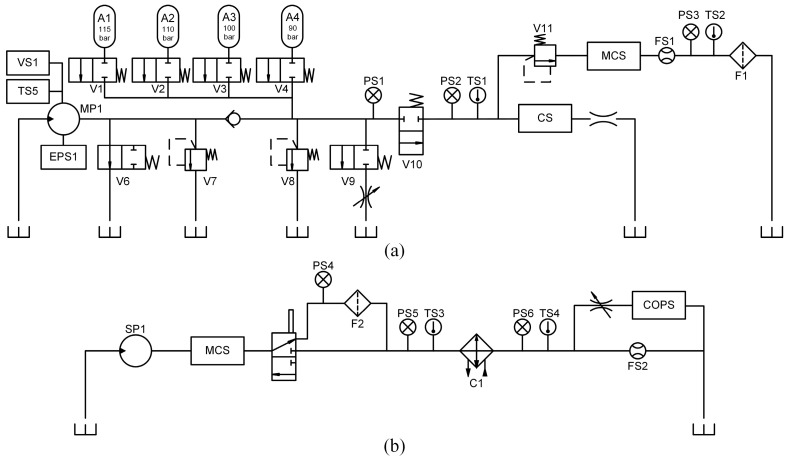
Hydraulic test rig in case II: (**a**) primary working circuit and (**b**) secondary cooling filtration circuit.

**Figure 8 sensors-24-00353-f008:**
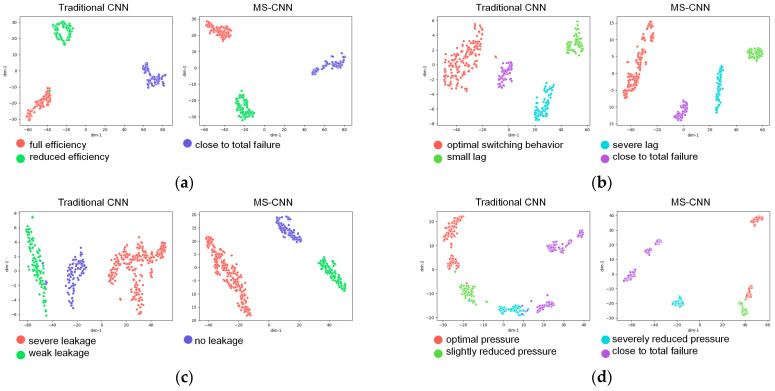
Feature visualization via t−SNE reduced from fault features for the testing dataset with (**a**) cooler, (**b**) valve, (**c**) pump, and (**d**) accumulator.

**Table 1 sensors-24-00353-t001:** Sensors in the hydraulic system.

Sensor	Description	Unit	Range
DP1-2	pressure of filters	bar	0 to 50
FS1-5	flow rate	L/min	−30 to 30
PS1-6	pressure	bar	0 to 345
TS1-3	temperature	°C	0 to 50
VC1-2	current of coils	mA	0 to 20
VV1-2	voltage of coils	V	0 to 50

**Table 2 sensors-24-00353-t002:** Hydraulic faults in the test rig.

Fault Label	Fault Condition	Emulation Parameters	States	Samples
LK1	Flow leakage in quick coupling (QC3)	Throttles of the flow restrictor (RV6)	0%: No leakage	200
			5–10%: Weak leakage	
			10–15%: Severe leakage	
LK2	Flow leakage in quick coupling (QC1)	Throttles of the flow restrictor (RV1)	0%: No leakage	200
			5–10%: Weak leakage	
			10–15%: Severe leakage	
LK3	Flow leakage in quick coupling (QC2)	Throttles of the flow restrictor (RV2)	0%: No leakage	200
			5–10%: Weak leakage	
			10–15%: Severe leakage	
BLK1	Flow blockage in quick coupling (QC3)	Throttles of the flow restrictor (RV5)	100%: No blockage	200
			90–95%: Weak blockage	
			85–90%: Severe blockage	
BLK2	Flow blockage in filter (F3)	Throttles of the flow restrictor (RV3)	100%: No blockage	200
			90–95%: Weak blockage	
			85–90%: Severe blockage	
BLK3	Flow blockage in quick coupling (QC4)	Throttles of the flow restrictor (RV4)	100%: No blockage	200
			90–95%: Weak blockage	
			85–90%: Severe blockage	
GL1	Gas leakage in accumulator (A1)	Pre-charge pressure of accumulator (A1)	80 bar: Optimal pressure	200
			70–75 bar: Slightly reduced	
			65–70 bar: Severely reduced	
GL2	Gas leakage in accumulator (A2)	Pre-charge pressure of accumulator (A2)	80 bar: Optimal pressure	200
			70–75 bar: Slightly reduced	
			65–70 bar: Severely reduced	
ACF	Fault in actuator (AC1)	Stroke length of actuator cylinder (AC1)	30 cm: No fault	200
			0–29 cm: Actuator fault	

**Table 3 sensors-24-00353-t003:** Layer configurations of the MS-CNN models.

Layer Name	CNN Models
CL-1 (Full-input-layer)	Conv (${29\times N}_{F},16$)
CL-1 (Partial-input-layer)	Conv (${29\times N}_{P},16$)
PL-1	Average pool (9)
CL-2	Conv (29×16,32)
PL-2	Average pool (9)
CL-3	Conv (29×32,64)
GMP	Global max pool
FC	Fully connected (${64\times N}_{Classes}$)
CL-1 (Full-input layer)	Conv (${29\times N}_{F},16$)

**Table 4 sensors-24-00353-t004:** Hydraulic system fault diagnosis performances of the proposed method with other algorithms.

	Method	Accuracy
1	PCA + SVM	70.15 ± 4.54%
2	PCA + ANN	78.70 ± 3.41%
3	DT + SVM	87.60 ± 3.15%
4	DT + ANN	94.10 ± 2.06%
5	GA + ANN	85.65 ± 2.84%
6	GA + RF	86.40 ± 2.39%
7	RF	88.65 ± 2.58%
8	ANN	80.65 ± 2.60%
9	MS-CNN	99.40 ± 0.81%

**Table 5 sensors-24-00353-t005:** Diagnosis performances of the proposed method with other CNN-based multi-sensor methods.

	Train Accuracy	Test Accuracy	Time Cost (s)	
			Train	Test
MS-CNN	99.88 ± 0.14%	99.40 ± 0.81%	56.0	0.27
CNN-1	89.56 ± 0.40%	89.55 ± 0.96%	803.8	0.64
CNN-2	99.56 ± 0.66%	98.30 ± 1.38%	392.6	0.39

**Table 6 sensors-24-00353-t006:** Hydraulic components and faults simulation methods in Case II.

Component	Condition	Value	Interpretation
Cooler C1	Cooling power decrease (%)	100	full efficiency
		20	reduced efficiency
		3	close to total failure
Valve V10	Switching degradation (%)	100	optimal switching behavior
		90	small lag
		80	severe lag
		73	close to total failure
Pump MP1	Internal leakage (code)	2	severe leakage
		1	weak leakage
		0	no leakage
Accumulator A1−A4	Gas leakage (bar)	130	optimal pressure
		115	slightly reduced pressure
		100	severely reduced pressure
		90	close to total failure

**Table 7 sensors-24-00353-t007:** Sensors in hydraulic system in Case II.

Sensor	Physical Quantity	Unit	Sampling Rate
PS1−PS6	Pressure	bar	100 Hz
EPS1	Motor power	W	100 Hz
FS1−FS2	Volume flow	L/min	10 Hz
TS1−TS4	Temperature	°C	1 Hz
VS1	Vibration	mm/s	1 Hz
CE	Cooling efficiency (virtual)		1 Hz
CP	Cooling power (virtual)	kW	1 Hz
SE	System efficiency factor (virtual)	%	1 Hz

**Table 8 sensors-24-00353-t008:** Result of the comparison with published studies and the traditional CNN model.

Sensor	Method	Accuracy (%)				
		Cooler	Valve	Pump	Accumulator	Average
Our proposed	MS-CNN	100.00	100.00	99.94	99.96	99.98
	Traditional CNN	99.85	100.00	99.85	98.95	99.66
Helwig et al. [40]	LDA	100.00	100.00	98.00	90.40	97.10
Liu [41]	LDA + HKELM	99.94	99.98	99.91	98.31	99.54
Wang et al. [42]	PCC + NCA + Deep Learning	99.91	100.00	99.77	98.70	99.60
Zhang et al. [27]	Deep transfer leaning	100.00	100.00	98.20	96.40	98.65
Huang et al. [26]	Multi-rate CNN	100.00	100.00	98.98	99.35	99.58

**Table 9 sensors-24-00353-t009:** MS-CNN-selected sensor channels for different faults in Case II.

Faults	Selected Sensor Channels
Cooler C1	[CE, TS4, PS5, CP, PS6, TS1]
Valve V10	[PS2, PS1, EPS1, PS3, FS1, SE]
Pump MP1	[SE, FS1, PS3, EPS1, PS1, TS2]
Accumulator A1−A4	[PS3, FS1, FS2, TS1, TS4, SE]

## Data Availability

Due to the nature of this research, participants of this study did not agree for their data to be shared publicly, so supporting data are not available.

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
