# Peer review of "Real-Time Fault Diagnosis for Hydraulic System Based on Multi-Sensor Convolutional Neural Network"

_sensors, 2024, doi:10.3390/s24020353_

Round 1
Reviewer 1 Report
Comments and Suggestions for Authors
In my opinion, the presented article is unreliable in its current form. To improve its quality and confirm the described content, the authors need to explain several aspects:
- the effectiveness of the method is over 99%, which is incredibly high efficiency considering that a significant part of the samples are removed from the signal. The Authors must show the nature of the waveform and the "healthy" signal parameters (e.g. amplitude, peak-to-peak, RMS) with the signal containing damage. Then, it will be possible to determine whether the method is effective for emerging damage (when the signal amplitudes are slightly different) or whether serious damage must occur (when the RMS parameter increases).
- formula no. 3, the equations should be swapped so that the description is the same throughout the article
- what does the order of the methods described in Table 4 mean? Why do the waveforms and values in the legend in graph no. 5 not match the table?
- why are only three methods compared in Table 5?
- why are different methods compared in Tables 4 and 8?
- line 461: why is it written that DT+ANN has the best performance? The table lacks, e.g. conversion time to prove it
- was drawing two drawn correctly?
- the literature review can be supplemented with:
a) line 161: Frequency and mode shape evaluation of steam turbine blades using the metal magnetic memory method and vibration wave propagation
b) line 138: Method of shaft crack detection based on squared gain of vibration amplitude
- the vast majority of the literature cited is from Asia. Please expand to other continents
- please correct the formatting of the literature. Some items include „et al.” others list all authors.
Author Response
Thank you for your nice comments on our article. As you are concerned, there are several problems that need to be addressed. According to your nice suggestions, we have made extensive modifications to our manuscript. The detailed corrections are listed in the attachment.

Reviewer 2 Report
Comments and Suggestions for Authors
This paper proposed a fault diagnosis method based on the MS-CNN that incorporates feature extraction, sensor selection, and fault diagnosis into an end-to-end model. From the experimental analysis, the proposed method could improve the abstract features extraction ability of deep learning model, comparing with several manually design based fault diagnostic methods. The subject is interesting, however, there are several address need to be improved.
(1) In abstract, the novelty and contribution of the paper need to be enhanced.
(2) What is the difference between the CNN in the proposed MS-CNN method and other CNN?
(3) In Section VI, a comparison is made between the proposed method and S-Transform and STFT to validate the use of MSST. How about other current time-frequency techniques, like WT, EMD ?
(4) What are the reduced sensing channels ? Is this universal for hydraulic systems ?
(5) It is suggested to add several fault diagnosis article, such as: 10.1504/IJHM.2022.125090
Author Response
We feel great thanks for your professional review work on our article. As you are concerned, there are several problems that need to be addressed. According to your nice suggestions, we have made extensive corrections to our previous draft, the detailed corrections are listed in the attachment.

Round 2
Reviewer 1 Report
Comments and Suggestions for Authors
The Authors clarified the reviewer's doubts and answered all his questions. Relevant explanations have been updated in the article. The article can be accepted in its present form.
Author Response
We would like to express our sincere thanks to the reviewers for your constructive and positive comments.
Reviewer 2 Report
Comments and Suggestions for Authors
Authors have improved the quality of the manuscript according to the comments, and the novelty and contribution are enhanced significantly. I think it can be published in Sensors.
Author Response
Thank you very much. We are very grateful to your effort reviewing our paper and your positive feedback.